# Long time-scales in primate amygdala neurons support aversive learning

Aryeh H. Taub [1], Yosef Shohat[1] & Rony Paz[1]

Associative learning forms when there is temporal relationship between a stimulus and a reinforcer, yet the inter-trial-interval (ITI), which is usually much longer than the stimulus-reinforcer-interval, contributes to learning-rate and memory strength. The neural mechanisms that enable maintenance of time between trials remain unknown, and it is unclear if the amygdala can support time scales at the order of dozens of seconds. We show that the ITI indeed modulates rate and strength of aversive-learning, and that single-units in the primate amygdala and dorsal-anterior-cingulate-cortex signal confined periods within the ITI, strengthen this coding during acquisition of aversive-associations, and diminish during extinction. Additionally, pairs of amygdala-cingulate neurons synchronize during specific periods suggesting a shared circuit that maintains the long temporal gap. The results extend the known roles of this circuit and suggest a mechanism that maintains trial-structure and temporal-contingencies for learning.

[1] Department of Neurobiology, Weizmann Institute of Science, Rehovot 76100, Israel. Correspondence and requests for materials should be addressed to R.P. (email: rony.paz@weizmann.ac.il)

In associative learning, the passage of time between trials—the inter-trial interval (ITI), can potentially serve as a cue of trial expectation. The time that passes from the offset of one trial carries information about the onset of the next trial, and if this time can be internally represented and kept, it can aid to form higher-order representations of the environment. Evidence that the ITI is indeed used for timing and auto-shaping of responses was first described in a series of classical work[1,2]. Importantly, the ITI length affects acquisition rate and memory strength[3–5], and the ratio between the ITI and the inter-stimulus interval (ISI) predicts acquisition characteristics[6,7]. This suggests that acquisition of associative memory involves assessment and integration of multiple temporal contingencies, and specifically, that temporal information about the ITI is internally represented even in the absence of behavioral changes during it.

Because the ITI usually lasts from few seconds to dozens of seconds, it raises the question where is this temporal lag being maintained? Neurons in the basolateral complex of the amygdala (BLA) play a role in acquisition and expression of affective associations[8–10], and show tonic responses and baseline changes that last after a specific trial or stimulus ends[11–13]. The amygdala also plays a role in the acquisition of trace-conditioning, where a temporal gap between the conditioned stimulus (CS) and the unconditioned stimulus (US) must be bridged[14–17], and even in longer lags that were traditionally thought to be dependent on the hippocampus[18,19]. In addition, the amygdala plays a role in second-order conditioning[20–22], which can be formed between the ITI itself and the CS[23] and even hold abstract representations of the trial structure beyond the CS-US relationship[24,25]. Further, the amygdala can use its bidirectional connectivity with the anterior cingulate cortex (ACC)[26]. The ACC is required for trace-conditioning with long gaps[27] likely due to its role in attention[27–29]. It forms and integrates representations of task structure[29,30], and exchange information with the amygdala for flexible updating of contingencies[10,16,17,31].

We therefore hypothesized that the BLA together with the ACC hold a representation of long timescales during the ITI of an aversive conditioning task, even in the absence of an explicit external cue. To address this, we recorded the activity of single neurons in the amygdala and the dorsal ACC (dACC) in non-human primates during a tone-odor conditioning paradigm with a relatively long (dozens of seconds) ITI that varies on a trial-by-trial basis. Our results show that longer ITI duration increases rate and strength of aversive learning, and that single units in the primate amygdala and dACC develop and hold a temporal representation of this long temporal gap, both in single-cell activity and at the population level. As a result, this circuitry maintains timescales at the order of dozens of seconds and can contribute to the formation of trial-structure and modulate affective learning and memory.

## Results

We recorded neuronal activity during the ITI in a tone (CS)-odor (US) conditioning task ($n_{days} = 82$), where an aversive odor was paired with a pure-frequency tone (250 ms, randomly chosen each session from 900 to 2400 Hz). Tones were triggered by onsets of respiration cycle detected in real-time, and odors were presented at the onset of the following respiration cycle (mean ISI duration of $1.6 \pm 0.24$ s). As shown in our previous studies[16,32]

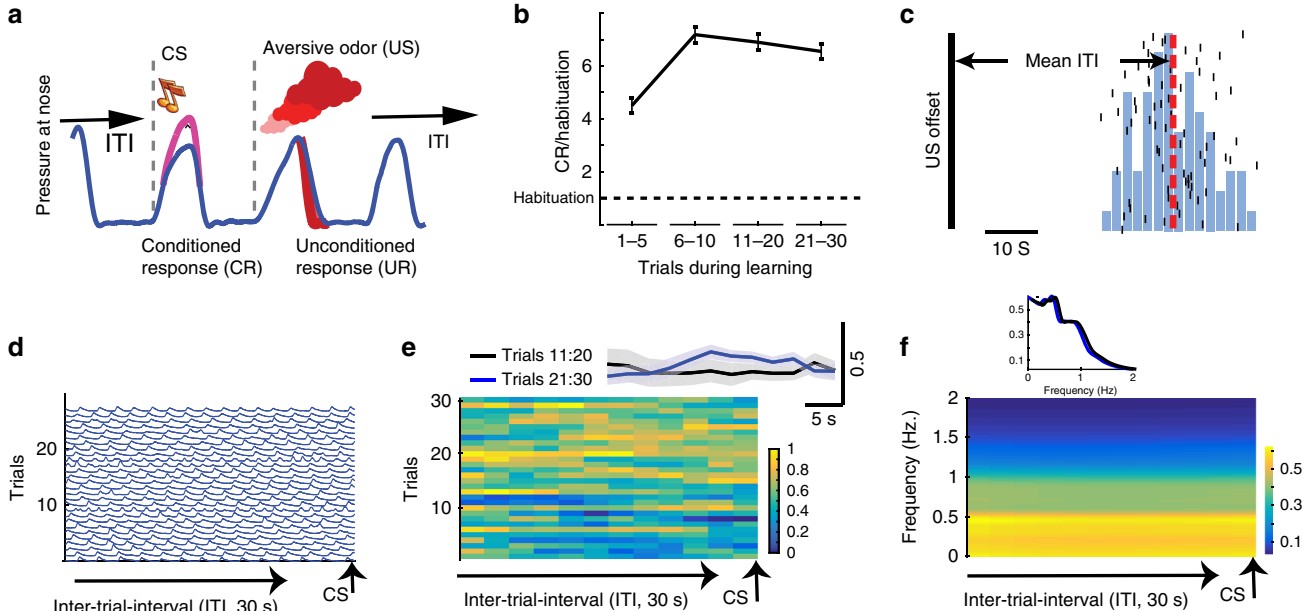

**Fig. 1** Conditioning paradigm and behavior during the inter-trial interval (ITI). **a** A trial scheme: online detection of inhale is used to initiate tone (CS) presentation that is followed by an aversive odor (US) at the onset of the next inhale. Response to the aversive odor is a decreased inhale (UR), whereas the conditioned response (CR) is an increased inhale. **b** The CR increases during learning. Shown is the CR (inhale size during the 350 ms post CS onset) as proportion of the inhale size during habituation (dashed line). Data presented as mean ± S.E.M. **c** A histogram of ITI duration from one representative session. Trial occurrences (black vertical line signals US offset) and the initiation of the subsequent trial (black vertical ticks). Dashed red line is the mean ITI at 42 s. **d** Inhale traces during the ITI aligned to onset of next trial. Shown are data for one session of learning. No clear modulation can be observed toward the end of the ITI. **e** Inhale modulation during the ITI did not change across learning. Shown is mean ± S.E.M across all learning sessions for the 30 learning trials (one-way ANOVA, $p > 0.1$, df = 11, $f = 0.05$). Inset presents mean over trials 11–20 and 21–30, computed during 30 s of ITI. **f** No change in inhale frequency across learning trials. Shown is mean power across all learning sessions at each frequency (0.2–2 Hz) during the ITI (one-way ANOVA, $p < 0.01$, df = 11, $f = 0.05$). Inset presents inhale frequency (mean over days, $n_{days} = 82$) as power during the whole 30 s of ITI preceding the CS, separately for trials 11–20 and 21–30

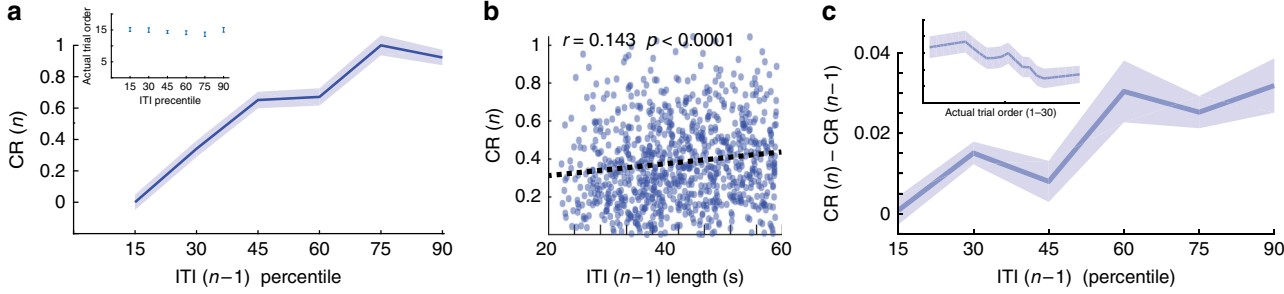

**Fig. 2** Length of ITI contributes to learning rate. **a** The size of the CR in a trial ($n$) as a function of ITI duration in the previous trial ($n − 1$), averaged across all sessions and binned into six percentiles (one-way ANOVA, significant effect for percentile, df = 5, $f$ = 29.85, $p < 0.0001$). Inset shows the averaged trial order sorted similarly by ITI duration percentile ($n$ = 171, one-way ANOVA, df = 5, $f$ = 0.74, $p$ = 0.59). **b** Furthermore, CR size was positively correlated with the previous ITI duration (Spearman correlation, $r$ = 0.14, $p < 0.0001$) on a trial-by-trial basis. **c** Change in CR amplitude between successive trials (CR in trial ($n$) minus CR in trial ($n − 1$) increase with ITI duration (one-way ANOVA, df = 5, $f$ = 81.14, $p < 0.0001$). ITI duration ($x$-axis) computed similarly to **a**. For control, inset shows the standard expected learning effect i.e. the same change in CR ($n$) − CR ($n − 1$) as a function of trials in learning. Data presented as mean ± S.E.M ($n_{days}$ = 82)

and here (Fig. 1a, b), pairing resulted in a CS-evoked augmented inhale that preceded the presentation of the odor, and reflects a learned preparatory conditioned response (CR).

**ITI length contributes to learning rate**. As might be expected from the long ITI duration and its trial-by-trial variability (mean ITI 38.7 ± 4.4 s, Fig. 1c), and from the fact that the following CS completely predicted the US, we did not observe any preparatory behavioral response during the ITI (Fig. 1d), neither in the depth of inhale modulation (Fig. 1e) nor in its frequency (Fig. 1f).

However, when sorting the CR based on the duration of the previous trial ITI, we found that the length of the ITI induced a higher CR, both in absolute value compared to habituation (Fig. 2a, b), and in increased learning rate (Fig. 2c). These results show that longer ITIs contribute to and enhance the learned behavioral response.

**Neurons modulate their activity during the ITI**. We recorded single-unit activity in the amygdala and the dACC (Fig. 3a, amygdala: $n$ = 291; dACC: $n$ = 263), and first validated previous results by comparing pre-CS to post-CS activity (paired $t$-tests, $p < 0.05$). We found that 24% ($n$ = 70) of amygdala and 29% ($n$ = 76) of dACC neurons had CS-evoked activity, confirming learning-related changes in these regions ($p < 0.001$, binomial test)[16,33].

We next turned to examine the activity during the ITI. On average, no specific epoch of the ITI showed modulation of activity (Fig. 3b). However, close inspection of the data suggested that many of the neurons do modulate their firing rates during the ITI, but over specific temporal scales (Fig. 3c). To test this, we analyzed the neuronal activity between the offset of each trial and the expected initiation of the next trial, namely the average ITI length (mean ITI), computed separately for each session. We omitted the bins where the CS occurred before the mean ITI to avoid CS-evoked activity in the analyses. Neurons' discharge during the ITI was binned into segments, and the strength of the ITI modulation was assessed by comparing the original ITI firing rate (ITI-FR) in each bin to shuffled data (Fig. 3d). We found that the activity of roughly 50% of amygdala and dACC cells differed from baseline activity for 20% or more of the total duration of the ITI (Fig. 3e, $p < 0.01$, binomial test). For robustness, we tested and observed identical results under different segmentations (6, 12, 20, and 40 bins, Fig. 3d, e). For comparison, a null distribution derived from shuffled data showed that almost all cells have significant activity in <5% of ITI duration, as expected from chance-level modulation (Fig. 3e).

We conclude that in both structures, large and similar proportions ($p > 0.1$, binomial tests, Supplementary Fig. 1) of cells had significant modulation during the ITI.

**ITI modulation strengthens during conditioning**. If indeed ITI modulation contributes to the acquisition, then we can expect it to strengthen during learning. To evaluate the strength of the ITI-FR modulation we computed the excess in spikes after normalization to the mean and standard deviation of the shuffled data (Fig. 4a, b). The root mean square (RMS) of 21% ($n$ = 61) of amygdala neurons and 26% ($n$ = 68) of dACC neurons was increased compared to habituation in one or more stages of the acquisition (early, mid, or late, 10 trials each; one-way analysis of variance (ANOVA), amygdala: df = 60, $f$ = 2.94, $p$ = 0.04 and ACC: df = 67, $f$ = 2.81, $p$ = 0.046). These acquisition responsive neurons (23%, 129/554) were further inspected during same-day extinction that started 15–30 min after the acquisition ended. Their ITI modulation diminished significantly yet remained higher than habituation (one-way ANOVA, amygdala: df = 60, $f$ = 3.66, $p$ = 0.0098 and ACC: df = 67, $f$ = 3.71, $p$ = 0.008, Fig. 4b). This result suggests that during acquisition of emotional memory, the activity of neurons in the amygdala and the dACC becomes locked to cycles of CS-US and the long ITI (dozens of seconds) between them.

Although odor exposure can remain in the receptors for few seconds and even induce prolonged behavioral effect, it cannot account for the ITI modulation we observe. First, the neural modulation often occurred during the middle or late segments of the ITI (see also next sections), yet to completely exclude this possibility we introduced "catch" trials with un-reinforced CS during the acquisition (<1/3 of the trials, during half of the sessions). We found that 11% (29/247; $p < 0.05$, binomial test) of the cells had significant modulation in the ITI following un-reinforced trials (the lower number of cells can stem from the lower number of trials used for analysis, hence lower statistical power). Importantly, the ITI response of these cells was highly similar in reinforced and un-reinforced trials (Fig. 4c, d, $t$-test, $p > 0.1$). Here again, their response diminished during extinction (Fig. 4d; $t$-test, $p < 0.05$). In sessions that included both aversive and appetitive odors conditioned to two different tones (discrimination learning), we found that the proportion of ITI-FR modulation was significant after both types of valence, but higher following aversive trials (Supplementary Fig. 1).

Overall, the findings are in line with the hypothesis that the FR modulations during the ITI occur during learning.

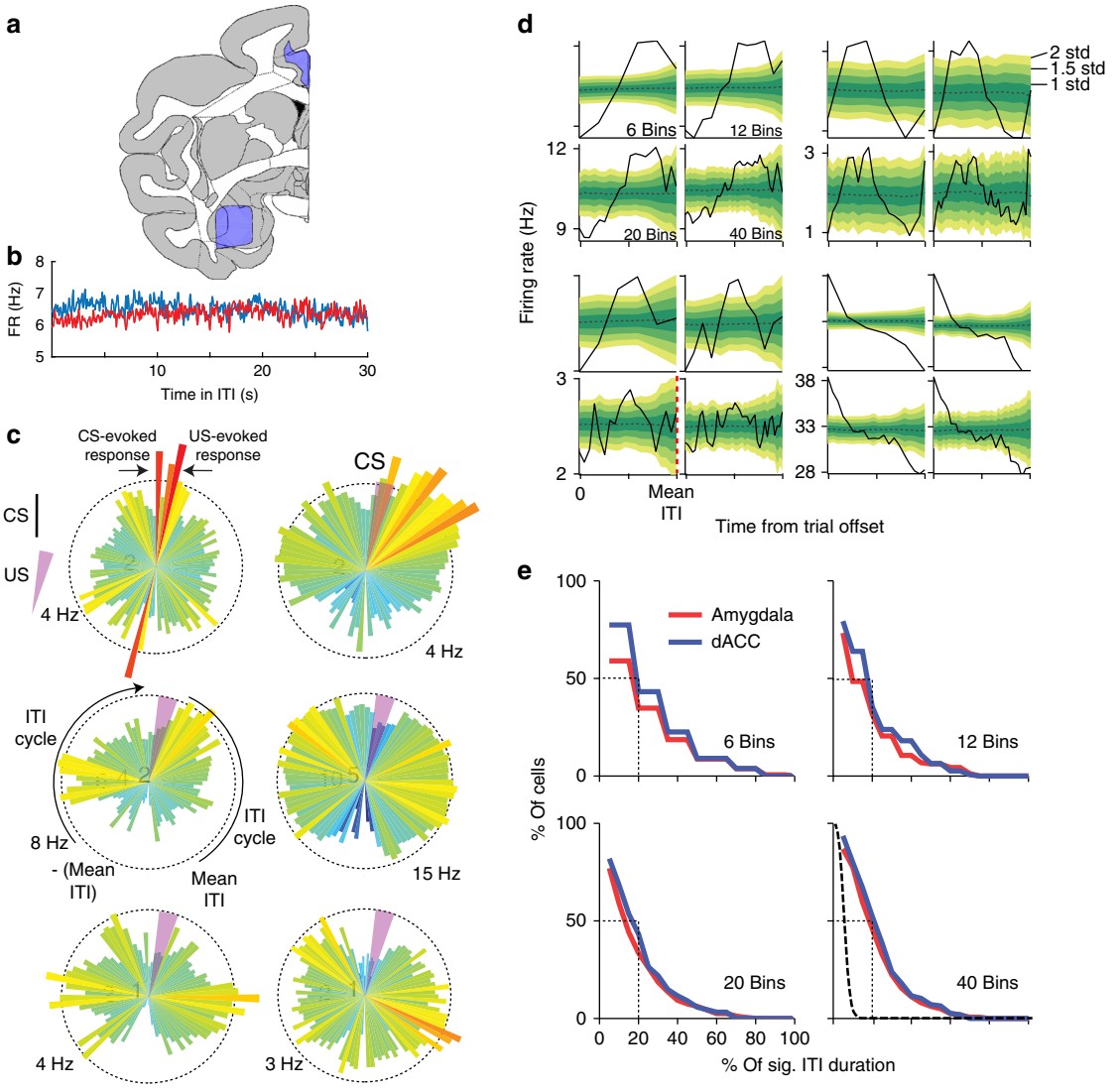

**Fig. 3** Neurons in the amygdala and the dACC are modulated during the ITI. **a** Recording locations were all made within the purple surfaces, shown on top of a macaque atlas (at AC = 0, IA = +20 mm, adapted from ref. [70] and http://www.braininfo.org. **b** Firing rate (FR) during 30 s aligned to next trial CS, averaged over all sessions (mean ± S.E.M) and neurons in both amygdala and dACC. No period showed specific modulation. **c** Single-cell firing rates of three amygdala neurons (left column) and three dACC neurons (right column). Modulation occurs not only in response to the CS and the US (as in the left top example) but also during different periods in the ITI. Shown are PSTHs for two ITI cycles, before and after the CS-US (CS is marked by black vertical line upwards, US by light purple wedge, and mean ITI in the session is marked in dashed red lines). The ITI that comes before the CS is on the left half-circle, and the ITI that comes after the US is on the right half-circle. The instantaneous firing rate is represented by the vector size and the colormap. **d** ITI-FR modulation evaluated from trial offset (time 0) to the mean ITI (dashed red line). Shown are four single-cell examples of the actual ITI-FR (black line) binned into 6, 12, 20, or 40 bins (for robustness) and the shuffled confidence interval (yellow-green shades). Notice that data were shuffled only between bins that occur before a new trial initiates, and as a result the confidence interval increases toward the end of the ITI. **e** Histograms of the proportion of cells (y-axis) that showed significant ITI-FR modulation (ITI-FR > 2 s.t.d. of the shuffled FR) in a proportion of the ITI duration (x-axis). For example, 50% of the cells had significant modulation in 20% or more of the ITI duration (dashed black line). All binning options are shown for robustness. A null distribution (obtained from shuffling) is shown for comparison in dashed black line, where most cells had significant modulation at around 5% of the time (as expected from chance). Single units: amygdala: $n = 291$; dACC: $n = 263$

**Trial-by-trial modulation of ITI signaling**. If indeed expectation is formed based on the mean of the ITI, and ITI length contributes to learning rate (Fig. 2), one can hypothesize that slight variations in length can induce transient updates in representation. Specifically, the duration of the ITI in the previous trial is expected to correlate with the time of modulation during the following trial. To explore this possibility, correlations were calculated between the center of mass for the ITI-FR in the current trial and the duration of the ITI of the previous trial, for all available neurons. Indeed, 15% of the

neurons significantly correlated with the duration of the previous ITI (Fig. 5a, b). Correlations were significantly more common in dACC neurons than in amygdala neurons (17.4% and 10.6%, respectively, $p < 0.01$, $\chi^2$ test for independence), yet in both regions their prevalence exceeded chance level ($p < 0.05$, binomial test). Earlier ITIs (trials $n - 2$, $n - 3$, and $n - 4$) had only limited effect on the center of mass (7–9% of cells, $p > 0.05$), and a multiple regression model including $n - 1/2/3/4$ suggested that the combined duration affect only few cells (4%).

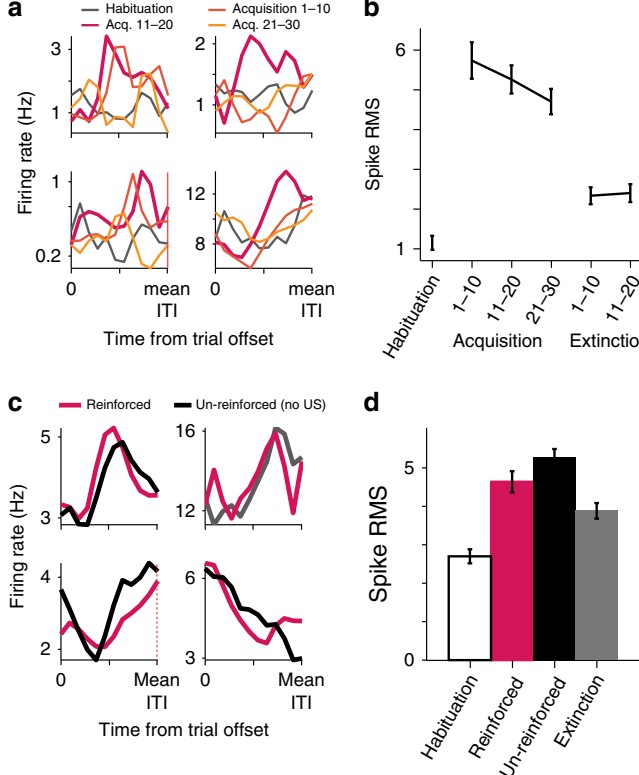

**Fig. 4** Modulation in ITI-FR increases during acquisition of emotional memory. **a** Shown are four examples of single cells with ITI-FR modulated more strongly during the acquisition (yellow-red shades) than during habituation. **b** The mean difference in spikes during the ITI increased in the acquisition compared to habituation (with slight decrease as acquisition progressed), and diminished partially during extinction. Shown is the average excess in spikes over shuffled data (quantified as RMS) across all cells with significant modulation. **c** ITI-FR modulations in un-reinforced trials (no US) was similar to the ITI-FR modulation in reinforced trials. Shown are four single-cell examples. **d** Cells with ITI-FR modulation in un-reinforced trials had similar modulation for the reinforced trials, and reduced ITI modulation in extinction (average over all cells with significant modulation during un-reinforced trials). Data presented as mean ± S.E.M. Single units: amygdala: $n = 291$; dACC: $n = 263$

This result demonstrates that neural activity during the ITI is constantly modified by the most recent trial.

To search for a more direct link between representation of time in neural activity and strength of conditioning, we correlated the behavioral change in CR (as in Fig. 2c) to the center of mass for the ITI-FR, for all trials and all neurons. This revealed a significant positive relationship (Fig. 5c, $r = 0.08$, $p < 0.0001$) that was slightly higher in the amygdala than in the dACC (Supplementary Fig. 2).

These results demonstrate that neurons adapt their temporal characteristics during the ITI based on the duration of the previously encountered period, and that this modulation is likely related to the behavioral change.

**Amygdala and dACC show peak modulations in different periods.** Our results so far suggest that amygdala and dACC neurons modulate their activity during the ITI, but is it a neural representation that spans the complete ITI duration? To address this we inspected single-cell peak modulation and found that 35% of amygdala cells (101/291) and 33% of dACC cells (89/263) exhibited a significant peak modulation ($p < 0.01$ for both, binomial tests).

Although the proportion of amygdala and dACC cells with significant modulation was highly similar ($p > 0.1$, binomial test), the distributions of peak locations were different (Fig. 6a, b, $p < 0.05$, permutation test comparing the regional distributions). Both regions had early peaks, likely US-related, but whereas amygdala neurons had an additional density of peaks around the middle of the ITI, additional dACC peaks occurred later in the ITI (Fig. 6a, b). Despite this, the mean width of individual tuning was similar in the amygdala and dACC (Fig. 6c left, $p > 0.01$, t-test), with no change in tuning width across conditioning trials (Fig. 6c, d). These results suggest that the amygdala and the dACC span the duration of the ITI, potentially enabling maintenance of long time periods in this shared network.

**Synchronized activity communicates temporal information.** If amygdala and dACC represent different types of information and time during the ITI, it is possible that they transfer and share such information to aid in the maintenance of long timescales. To examine this, we computed inter-regional correlations between pairs of amygdala and dACC neurons that were recorded simultaneously. The cross-correlations were computed in 3-s window that advanced in 1-s steps from the end of each trial.

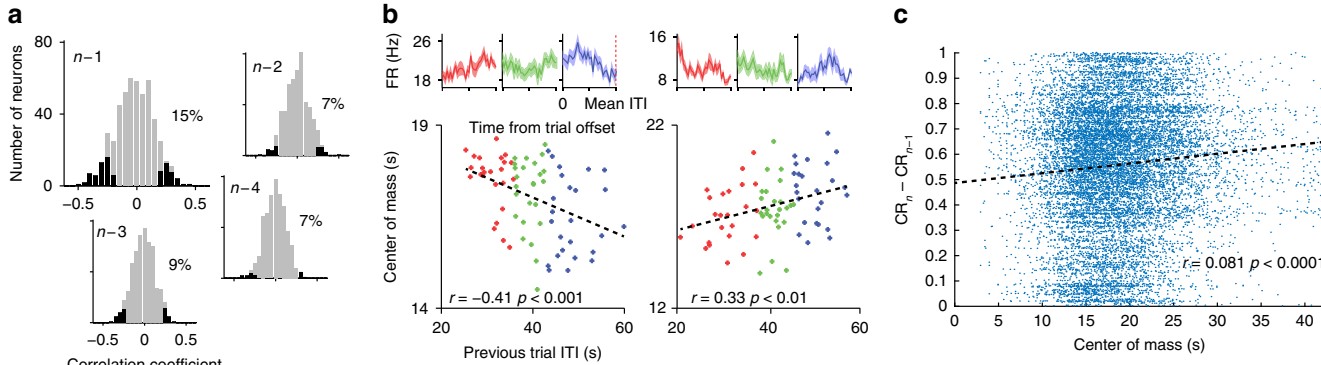

**Fig. 5** ITI duration affects the temporal-tuning in the next trial. **a** The location of the preferred time is correlated with the duration of the previous trial ITI in 15% of all cells ($p < 0.01$, binomial test). Earlier ITIs (trials $n − 2/3/4$) affect much less. **b** Two examples of cells in which the preferred time (quantified as center of mass, y-axis) correlates with the ITI duration of the previous trial (x-axis). The PSTHs of the lower third (red), middle third (green), and upper third (blue) are shown on top. Data presented as mean ± S.E.M. **c** Change in CR amplitude between successive trials plotted against the center of mass of neural activity in the ITI between the trials. Shown are all single units in all trials they were recorded, together with the linear fit. $n_{days} = 82$; single units: amygdala: $n = 291$; dACC: $n = 263$

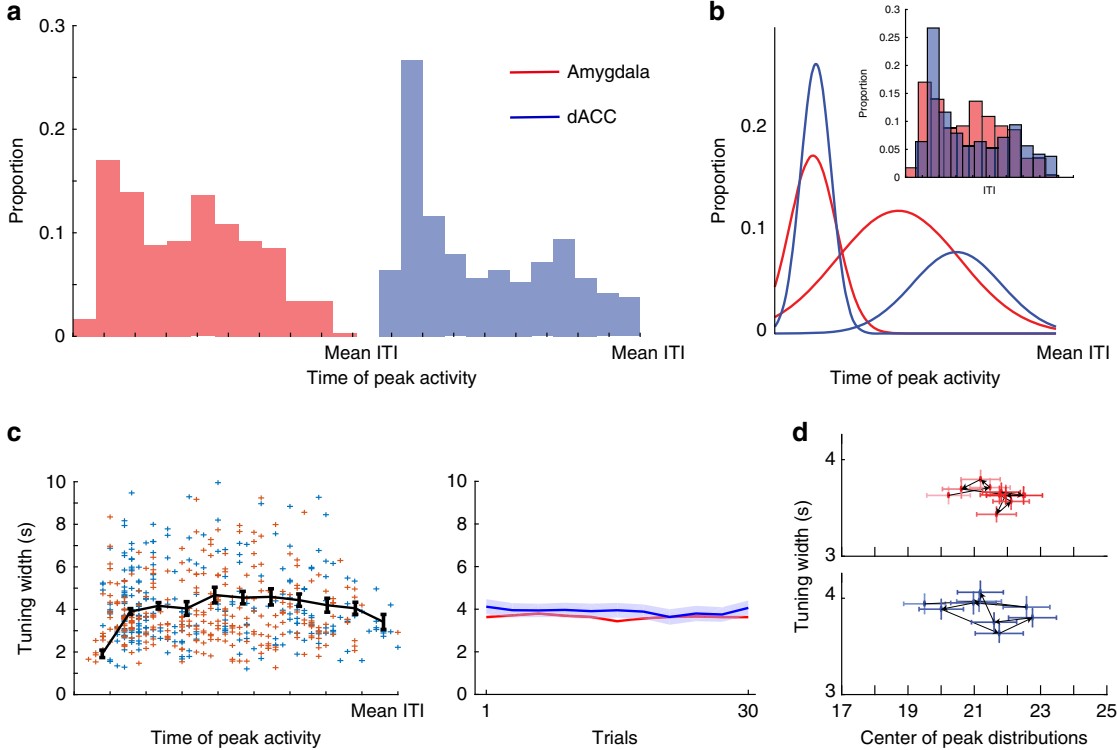

**Fig. 6** Peak activity during the ITI is differentially distributed. **a** Histograms of the locations of single-unit peak activity for both regions. **b** Fitted gaussians for the histograms shown in **a**; inset shows the two distributions overlaid. Amygdala: SSE = 0.0025, $r^2$ = 0.8111 and SSE = 0.0011, $r^2$ = 0.9238; dACC: SSE = 0.0152, $r^2$ = 0.5695 and SSE = 5.3466e-04, $r^2$ = 0.7565. **c** Left: tuning width of individual cells plotted against the time of peak activity for amygdala (red) and dACC (blue) cells, with mean ± S.E.M. (black). Middle: the average width did not change during acquisition (one-way ANOVA; amygdala: df = 10, f = 1.2, p > 0.28; dACC: df = 10, f = 1.78, p > 0.059; t-test for between-area difference, p > 0.07). **d** Mean tuning-width (from **c**) against mean peak location (from **a**) for all cells, shown for bins of five trials during acquisition (one-way ANOVA; amygdala: df = 10, f = 0.96, p > 0.46; dACC: df = 10, f = 1.58, p > 0.1; t-test for between-area difference, p < 0.001). Both parameters showed stability during learning. Single units: amygdala: n = 101; dACC: n = 89

Evaluation of significance in such cross-correlations requires two sets of shuffled data to address two different null hypotheses (Supplementary Fig. 3). First, data were shuffled within-ITI, so that spikes from a specific ITI were shuffled within itself, thus maintaining the overall firing rate in each ITI. This shuffle targets pseudo-correlations that may arise when the overall ITI-FR rate of the two units covaries. Second, data were shuffled across-ITI, so that spikes are shuffled between different trials' ITI—but from the same period within the ITIs. This addresses pseudo-correlations that may originate when the ITI-FR is repeatedly modulated along the ITI (Supplementary Fig. 3). Only cross-correlations that differed from both were deemed significant. A total of 41% (344/839) pairs of amygdala-dACC neurons were found to have a significant cross-correlation. Most correlations had a zero-lag with no clear physiological direction, suggesting reciprocal interaction between the amygdala and the dACC (Fig. 7a). However, the distribution of significant cross-correlations peaked mainly at early and late periods of the ITI (Fig. 7a).

We next examined if this double-peaked pattern reflects an average correlation map of prototypic pairs of neurons, or a mixture of single-peaked correlations. A principle component analysis (PCA) on the normalized distribution of correlation density along the ITI identified three separable clusters (Fig. 7b, c; k-means clustering). The distribution of cross-correlations of each cluster strengthens the previous finding and shows that it is comprised of different pairs that synchronize during a single and confined segment of time within the ITI. These segments tile the complete duration on one hand (Fig. 7c), but are much more dominant early and late during the ITI.

The results suggest that BLA-dACC neurons signal and potentially transfer temporal information in a pairwise-specific manner.

**Estimating time during the ITI with neural activity.** If neurons have temporal-tuning with peaks in different times during the ITI, and these are relatively homogenously distributed when combined across the amygdala and the dACC, then one should be able to decode time during the ITI (and as a result, its total length). We used an optimal linear estimator (OLE) with cross-validation and found that time could be estimated based on the population activity within a reasonable error (Fig. 8a), that dropped with the size of the population being used, yet remained within few (<4) seconds accuracy (Fig. 8b). Notice the true error is likely larger because we assume independence between neurons (similar results were obtained with a population vector approach and a Naïve Bayes classifier). Averaging the estimates showed that it is unbiased (Fig. 8c), and therefore a downstream network can use this information reliably to assess ITI duration. Supporting this, we used a hazard-rate approach and found that a proportion of the neurons had gradual elevated activity toward the end of the ITI, one that correlates with the increasing probability for CS appearance as the time in the ITI progresses (Fig. 9).

**Discussion**

We show here that neurons in the basolateral complex (BLA) of the primate amygdala together with neurons in the dACC acquire temporal-tuning modulations during the ITI. As a result, the population faithfully represents the time that passes between

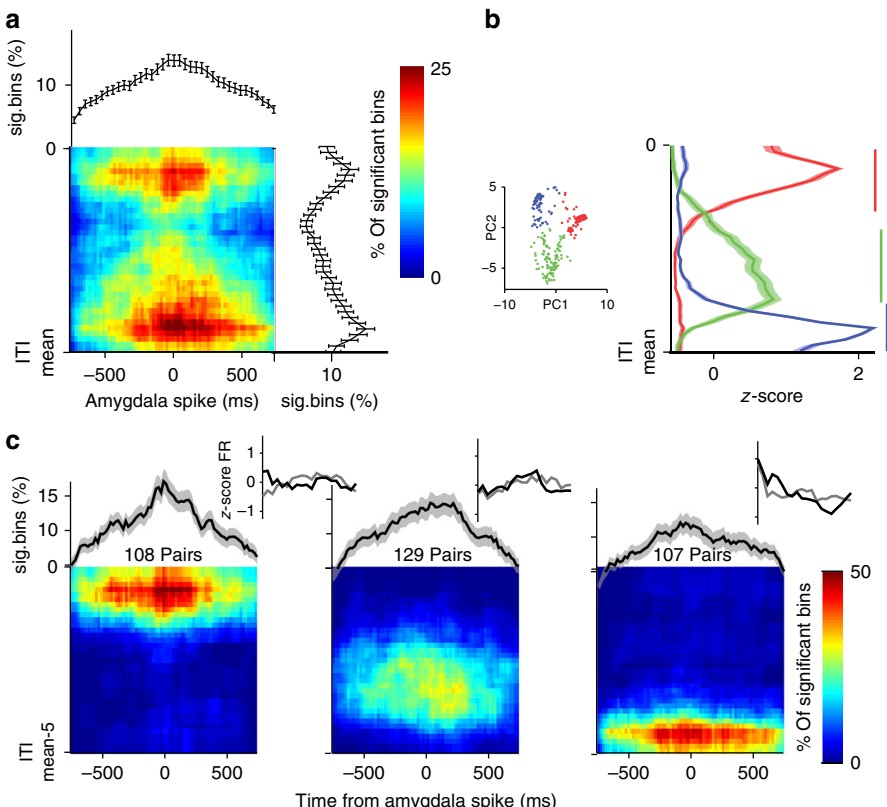

**Fig. 7** Amygdala-dACC pairwise-correlations peak at specific times during the ITI. **a** Cross-correlations were computed between simultaneously recorded amygdala-dACC neurons in sliding 3-s segments of the ITI. CCs were compared to shuffles to exclude changes resulting from covariations within and across ITIs (methods and Supplementary Fig. 3). Shown is the map of % significant bins from all pairs, for the duration of the ITI (y-axis, top to bottom) and triggered on amygdala spike (x-axis, time zero). Upper plot shows the average (marginal) CC over the whole ITI, and right plot shows the average (marginal) percentage of sig. bins along the ITI. **b** Shapes of individual CCs were clustered (inset, k-means on principal components after z-score), and the average shape for each cluster is presented; revealing two distinct times of CC peaks—early (red) and late (blue) in the ITI, and a third (green) lower widespread modulation. Horizontal colored bars indicate regions with significant synchronization (p < 0.01; t-tests). **c** Same as in **a** presented separately for each cluster found in **b**. Pairs with early (left) and late (right) CC were more focused in time during the ITI and exhibited stronger synchronization. Data presented as mean ± S.E.M. n = 344 pairs of amygdala-dACC neurons

trials, and can potentially represent anticipation of the aversive event that will be cued by the CS. In addition, because the length of the ITI contributes to learning rate and the strength of the memory, as shown here for aversive conditioning, our results suggest that this is mediated by amygdala-prefrontal networks. This is the first demonstration of representation of long timescales (dozens of seconds) in primate amygdala networks, an order of magnitude longer than the typical duration used for the ISIs. Therefore, this network can support formation and maintenance of temporal contingencies not only when they are cued explicitly by external stimuli, but also when embedded in the temporal structure of the task. Below we discuss the implications of this finding to valence-based learning.

Our finding of neural representation of time in BLA-dACC network could support several mechanisms that contribute to the effect of ITI duration on acquisition (learning) and CR (performance), as demonstrated across species and learning tasks[34–36]. Accordingly, we show here that the learning rate and size of the CR were positively correlated with ITI duration[3–7]. The underlying mechanisms for this effect can consist of multiple parallel processes.

One appealing explanation of ITI-related enhancement stems from the ratio between the ITI and the ISI (CS-US duration). Larger ratio is correlated with accelerated learning[5,7], in line with the larger informative power of the CS on the occurrence of the US[37]. Here we demonstrate one major component that of

time within the ITI. This is a more challenging aspect for a neural network and hence the main novelty in our study, because it requires representation of long timescales beyond that of persistent activity in single cells (as widely demonstrated in CS-US trace-conditioning, CS-delayed-response memory tasks, and interval-reproduction tasks). Nevertheless, full examination of this specific hypothesis would require longer ISI to allow the CS response to decay, and it would be intriguing to explore neural implementation of ratio-dependent acquisition rate. Here the ISI was short and predictable, and hence our finding of unbiased decoding error along the ITI can enable a representation of duration and ratio. Moreover, this interpretation is further supported by the finding that even single neurons shift the temporal characteristics of their activity based on the duration of the previous trial, and this shift was correlated with the instantaneous learning rate, hence providing a direct link between single-cell representation and behavior. Interestingly, we found a relationship only with the previous trial, suggesting that neurons use short-term history to update their expectation about ITI duration. This could be a result of the statistics used in the current study because the ITIs were normally distributed, and the mean duration can be computed additively on a trial-by-trial basis.

Despite the elegance of this temporal account of ITI duration and/or ITI-to-ISI ratio, several studies suggest that it might be insufficient. This is based on the observation that the proximity of

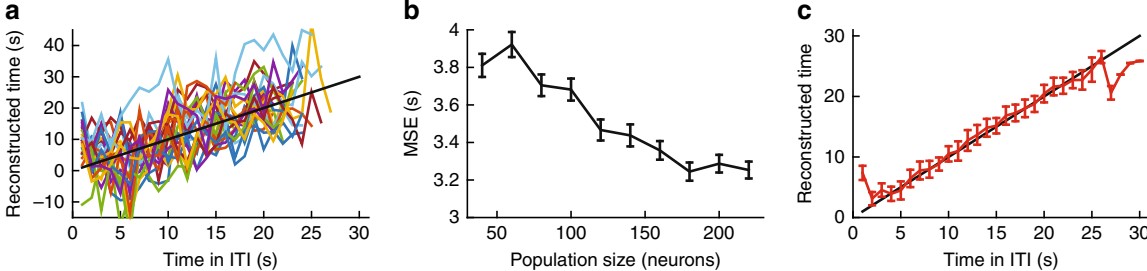

**Fig. 8** Time in ITI obtained from neural activity. **a** Decoding the time in the ITI from neural activity. Shown are 25 individual repetitions, when calculating each neuron preferred time on a set of 25 trials (drawn pseudorandomly for each repetition), and using it to estimate time from the actual firing rate in the remaining 5 trials. Black line marks identity line (perfect prediction). **b** The estimation error as a function of number of neurons used (neurons were taken as independent and collected from all days). The error drops, but seems to plateau around 3 s resolution. **c** Average of 100 repetitions using all available neurons. The population estimates the time in the ITI with small bias (edges are biased due to floor/ceiling effects). Black line marks identity line (perfect prediction). Data presented as mean ± S.E.M.

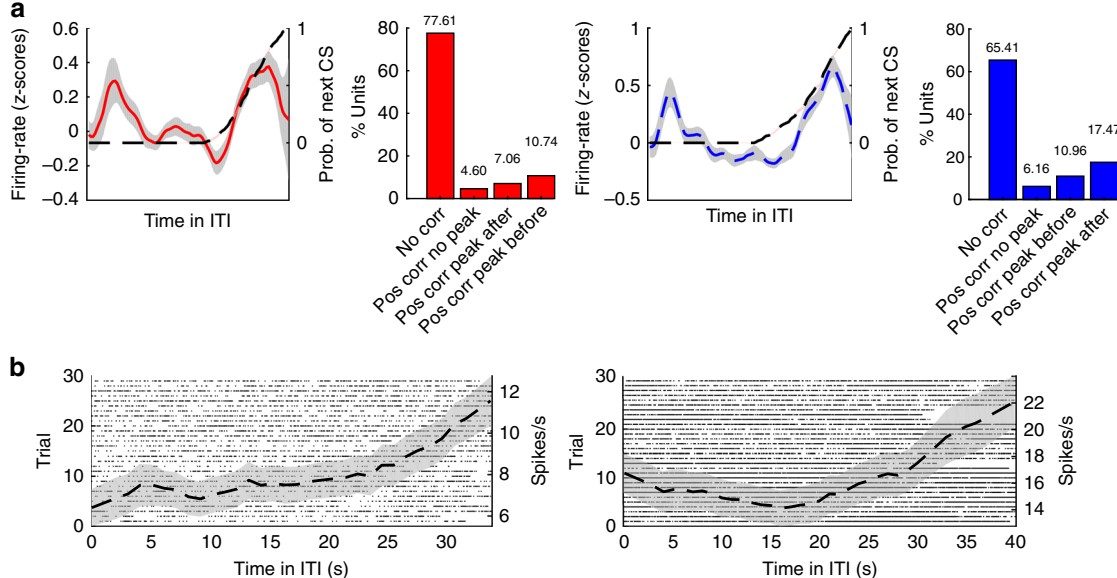

**Fig. 9** Activity at the end of the ITI. **a** Empirical hazard rate for the CS over all trials (black dashed line, right y-axis). A subpopulation of cells in the amygdala (left/red; 22.39%) and dACC (right/blue; 34.59%) showed a significant positive correlation with the rising phase of the daily hazard rate (shown is the mean z-scored firing rate with S.E.M., left y-axes). Within these significant hazard cells, the majority manifested a significant peak in activity during the ITI, either before the minimal ITI duration (amygdala 7%; dACC 10.96%) or following it (amygdala 10.74%; dACC 17.47%). Time and neural activity were warped for averaging and presentation purposes. **b** Rasters and PSTHs from amygdala (left) and dACC (right) single units that had a significant correlation with the daily hazard rate (up to 0.75, see methods). Data presented as mean ± S.E.M.

an ITI-interfering event to the preceding[38] or consecutive[39] trial increased the effect on learning performance. As alternative, the opponent process theory postulates that after an aversive US, an opponent relief-like process is initiated that interferes with the learning and longer ITI allows for this process to terminate and for learning to be enhanced[34,40]. Our finding that a larger proportion of BLA neurons are tuned to the middle of the ITI can support this, as the relief process evolves after the termination of the US yet decays toward the following trial, in line with safety signals reported in BLA neurons[41,42]. Further, the opponent process can also be activated by CS presentation alone as learning progresses, and reduced following extinction[40]. Accordingly, we found higher response magnitude during un-reinforced trials that diminish with extinction.

Another option is the formation of second-order associations mediated by the amygdala[20–22], first between the CS and the US, and then between the ITI itself and the CS, or between the US offset and the next CS (via the ITI "filler"[23]). However, we did not observe any behavioral changes during the ITI in

expectation of the next trial, and we therefore think it is a less probable interpretation. Even if such second-order association indeed takes place, it would require temporal activity as we report here that tiles and bridges the duration of the interval. Albeit, the ITI here was of varying random duration and hence the next CS cannot be faithfully predicted from US offset, yet the coming CS is a complete predictor for the US. Together with the lack of preparatory (CR-like) behavior during the ITI, this supports the interpretation that it is not a mediated association, but rather a representation of trial structure and/or statistics of the environment that is represented in amygdala networks[24].

In addition, behavioral[38] and neural[43,44] studies suggest that rehearsal of memory can occur during the ITI, where additional processing contributes and consolidates learning[45]. Although it is not clear at this point how representation of time and rehearsal are related and integrated, and evidence for neural rehearsal processes in the amygdala is still scarce, further characterization of BLA and dACC contribution to ITI-related processing could

shed light on these mechanisms and their putative complementary role.

The representation of long timescales on the order of dozens of seconds, and without an external cue as an anchor, is a new finding in the primate amygdala. One can consider several brain regions as candidates to supply this information to the amygdala. The striatum is one, shown to underlie and pace short timescales at hundreds of milliseconds[46], but also to signal longer durations[47,48], yet mostly in specific dedicated time tasks[35,49]. However, the striatum does not project directly to the BLA, yet the BLA does project to parts of the striatum[50,51]. In contrast, temporal information can reach the ACC via cortico and cortico-striatal loops[52–54], and from there to the amygdala[26]. Paralleling this, we observed a higher proportion of dACC neurons that scale their peak ITI modulation with the length of the recent ITI, on a trial-by-trial basis. Similarly, the representation of time can reach the amygdala via time-cells and event-integration in the hippocampus[55–57]. Whereas the exact functional role of such loops remains to be examined, it supports the idea that time perception is mediated by multiple overlapping neural systems, which are flexibly engaged depending on the task requirements[58]. In the current case, they can play a role in valence/emotional-based learning and support the demonstrated contribution of the ITI to memory strength.

An additional interpretation for the differential peaks of modulation observed in the two regions can come from functional considerations. The amygdala was shown to signal safety[41,42], in accordance with its view as an absolute valence decoder[59,60]. ACC neurons signal anxiety, threat, and risk[61,62], and importantly, the ACC plays a major role in error, context, and attention[28,29,63]. These functions are higher either right after or right before a trial. Therefore, the subtle distribution of roles might underlie the differential preference of ACC neurons to mediate information early and late in the interval, whereas amygdala neurons signal its middle.

Here we focused on the reciprocal functional loop between the BLA and the dACC and found that together they span the complete ITI duration. Moreover, we observed potential transfer of temporal information in confined periods, early and late, by zero-lag synchronization between simultaneously recorded pairs of neurons. We identified three classes of pairwise interactions that synchronize early in the ITI, late, and more diffused population that spans the middle of it. This is in line with other types of information that transfer between the amygdala and the dACC at the single-cell level[16,17,31,64–67] and orchestrated by field-oscillations[31,32].

These two seemingly different interpretations for the difference in peak distributions are not mutually exclusive and can serve both purposes. Namely, utilizing the tendency of one region to report a specific state aids to maintain the temporal marker for that state. This enable the network to use the underlying state of the animal to maintain additional information about the temporal structure across the network. Even if the immediate goal of the network is to represent the overall length of the ITI, it cannot do so by single-cell sustained activity, because of the high energetic demands for maintaining firing rates. The temporal-tuning-like approach allows this, and especially when distributed across regions.

To conclude, we find here that the activity of amygdala and dACC neurons hold information about long timescales, and together support learning rate and memory strength during aversive learning. The results provide a new role for this network in maintaining timescales to build the temporal and/or statistical structure of the environment, and suggests a mechanism to how longer intervals can promote learning and memory. In turn, it can also explain why under circumstances of multiple subsequent experiences, aversive memories are formed faster and exhibit strong-to-abnormal responses. Hence, in addition to existing learning models that implicate this specific network in anxiety and fear-disorders[9,10,68,69], our findings suggest a new model for how deviations in representation and computation in this circuitry can lead to maladaptive and exaggerated behaviors.

## Methods

**Animals.** Two male macaca fascicularis (5–7 kg) were implanted with a recording chamber (27 × 27 mm) above the amygdala and cingulate cortex under deep anesthesia and aseptic conditions. All surgical and experimental procedures were approved and conducted in accordance with the regulations of the Weizmann Institute Animal Care and Use Committee, following National Institutes of Health regulations and with Association for Assessment and Accreditation of Laboratory Animal Care (AAALAC) accreditation. Food, water, and enrichments (e.g. fruits and play instruments) were available ad libitum during the whole period, except before medical procedures that require deep anesthesia.

**Recordings.** The monkeys were seated in a dark room and each day 3–6 micro-electrodes (0.6–1.2 MΩ glass/narylene-coated tungsten, Alpha Omega, Israel or We-sense, Israel) were lowered inside a metal guide (Gauge 25xxtw, OD: 0.51 mm, ID: 0.41 mm, Cadence Inc., USA) into the brain using a head-tower and electrode-positioning system (Alpha Omega, Israel). The guide was lowered to penetrate the dura and stopped at 2–6 mm in the cortex. The electrodes were then moved independently further into the amygdala or the dACC. Electrode signals were pre-amplified, 0.3 Hz–6 kHz band-pass filtered and sampled at 25 kHz; and online spike sorting was performed using a template-based algorithm (Alpha Lab Pro, Alpha Omega, Israel). Anatomical magnetic resonance imaging scans were acquired before, during, and after the recording period and these scans were used to guide the positioning of the chamber on the skull at the surgery and to calibrate the positioning of the electrodes in the amygdala and the dACC.

**Behavior paradigm and stimuli.** Each daily session consisted of tone presentations (900–2400 Hz) that were triggered by real-time detection of respiration onset and followed by odor release at the onset of the following respiration cycle (but not earlier than 1 s from tone presentation). Odors (propionic acid or banana-melon organic extracts diluted in mineral oil) were actively evacuated from the nasal mask by a vacuum hose and respirations were constantly monitored with two parallel connected pressure sensors (1/4″ and 1″ H2O pressure range, AllSensors).

During the habituation phase the tones (novel tones on each day) were presented 10 times, without any odor to follow. During the acquisition stage, the tones were presented 30 times each, and paired with aversive odors in all occasions. In some sessions, aversive and pleasant trials were intermingled in a pseudorandom order and paired with different tones (discrimination learning). In addition, in half of the sessions, the acquisition stage also included 10 presentations of unpaired (catch) trials, i.e. tones that were not followed by an odor.

**Behavioral analysis.** Learning-dependent changes in the response to the CS were computed as area under the curve during 350 ms following CS onset compared with mean inhale volume in response to the same tone during habituation, as in previous work[16].

Behavioral ITI data were aligned to the end of the ITI, namely to the beginning of the next trial (CS occurrence), because our goal was to examine if any anticipatory/preparatory behavior develops toward the next trial. To evaluate changes in inhale power during the ITI across trials we averaged the area under the curve of breathing data during 30 s of the ITI with a running window of 7 s and overlap of 2 s. To evaluate changes in inhale frequency (for 0.2–2 Hz, steps of 0.01 Hz) we quantified spectrograms by obtaining the multitaper power spectral density estimation (Thomson multitaper method) for breathing data during 30 s of the ITI (7 s windows with and 2 s overlap).

**Neural data analysis.** Neural activity during the ITI was aligned to the end of each trial, and taken until the end of the mean ITI in each specific session, with exclusion of bins from trials with shorter ITI (to avoid sampling bias and CS response contaminating the dataset). This is because our hypothesis was that neural activity signals the time during the ITI that passes from the previous trial (as the reliable way to estimate the duration of the ITI).

*ITI-FR modulation:* Spikes discharge that occurred between the US offset and the mean ITI were binned into 6, 12, 20, or 40 bins, and counted to produce an estimation of the ITI-FR. To assess the strength of modulation, spikes were shuffled 100 times and the mean ITI-FR was compared to the mean and standard deviation of the shuffled ITI-FR distribution. Data were taken only from bins that occurred before the ITI-mean, to avoid sampling bias. Up and down modulations were determined when the original spike count at a specific bin exceed ±2 standard deviation of the shuffled spike count at the corresponding bin.

*Acquisition of ITI response:* ITI-FR was computed separately for habituation, acquisition, and extinction ITIs. The ITI-FR of the acquisition and extinction was

further separated into three and two equally sized (10 trials) sub-stages, respectively. Each ITI-FR was then normalized according to its own shuffled dataset (as above). This normalization assures that the ITI-FR at all stages will be on the same scale even in cases that the neurons increase or decrease their firing rate along the session (i.e. non-stationary). Next, the RMS of the ITI-FR was derived and compared between the different stages. ANOVA test was employed to examine whether the RMS scores changed between the habituation and the acquisition. When the ANOVA test indicated that the RMS was modulated during the session, we further performed post hoc *t*-test comparisons between the early, middle, and late acquisition phases and the habituation. Neurons whose ANOVA test was significant and with one or more post hoc comparisons that indicate that their RMS is increased during the acquisition stages were classified as acquisition responsive neurons.

*Peak modulation:* Spikes were binned into 40 bins during the ITI, and the bin with the highest firing rate (averaged over trials) was marked as the peak. This bin was then gradually joined by surrounding bins if they had a higher-than-baseline firing rate (with confidence interval > 95%). The number of bins were taken as the tuning-width (in seconds; units with tuning of 1 bin only were excluded).

*ITI correlations:* Pearson correlations between the center of mass of the ITI-FR and the length of the previous ITI (duration) were computed. Correlation at $p < 0.05$ were considered significant, and binomial test was employed to determine whether the percentage of units with significant correlation is higher than chance level (5%). Similarly, Pearson correlations were also computed between the RMS of individual ITI-FR during the acquisition and the length of the ITI at the previous trial.

*ITI cross-correlation:* Cross-correlations were computed between all pairs of amygdala-dACC that were recorded simultaneously and which firing rate exceed a mean of 1 Hz during the ITI (to allow sufficient reliable statistical power). Cross-correlations were computed in a 3-s window that was advanced in 1-s steps from the offset of the US until 5 s before the mean ITI. In all cases, amygdala spikes were used as the reference for the occurrence of dACC spikes, and spike that occur up to 750 ms before or after an amygdala's spike were included in our analysis. Next, spike times were binned into 20 ms bins and counted. To evaluate whether the observed cross-correlations were significant we repeated the same procedure but this time, the dACC spikes were shuffled in two different manners: (1) within-ITI shuffling, in which we shuffled spikes in each individual ITI, destroying their temporal resolution, but maintaining the overall firing rate in each individual ITI; and (2) across-ITI shuffling, in which spikes were shuffled between the different ITIs, but maintaining the same epoch within the ITI. This procedure changed the overall firing rate of individual ITIs, but maintains the temporal structure of the ITI-FR. Cross-correlations were deemed significant only if they exceeded both criteria, with a further imposed requirement that they maintain significance for at least 15% of the total ITI duration.

*Decoding time from neural activity:* We used cross-validation and an OLE. In each iteration, we pseudorandomly chose 25 trials from the acquisition and derived the optimal weights given the normalized firing rate in each bin of the actual time during the ITI (40 bins). These weights were then used to estimate/decode the time in each of the remaining 5 trials separately by using the real firing rates. We then averaged across neurons. The process was repeated 100 times, and also as a function of *N* neurons pseudorandomly selected from the different sessions (Fig. 8). A population vector approach gave highly identical results.

*Center of mass:* Center of mass was expressed in seconds and computed as

$$\text{Center of mass} = \frac{\sum_{1:n} \text{Firing Rate}(i) * \text{Time}(i)}{\sum_{1:n} \text{Fring Rate}(i)}$$

*Hazard rate:* Hazard rate was computed empirically for each session as the cumulative probability for CS occurrence in each time point, and normalized (warped) to the maximal ITI in each session for averaging across sessions. To identify matching neural activity, we correlated the activity of each cell firing rate with a portion of the hazard function, from the first point that rises from 0 to 0.75 (we used 0.75 because there only few trials that can be used for neural activity beyond it, inducing unreliable noise in the results). Firing rate was calculated in spikes/s in windows of 1 s advancing in steps of 0.5 s, normalized to the actual number of trials per ITI duration, and *z*-scored for purposes of averaging across units. Peak location was taken as in Fig. 6.

We repeated the main analyses separately for each monkey to validate the main results (Supplementary Fig. 5).

**Code availability**. Custom code for behavioral and electrophysiological tests is available from the corresponding author upon reasonable request.

## Data availability

All data supporting the findings of this study are available from the corresponding author upon reasonable request.

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

## Acknowledgements

We thank Drs. Rita Perets and Yoav Kfir for scientific advice; Dr. Uri Livneh for providing experimental data and analyses; Dr. Eilat Kahana for help with medical and surgical procedures; and Dr. Edna Furman-Haran, Nachum Stern, and Fanny Attar for MRI procedures. This work was supported by ISF #26613 and ERC-2016-CoG #724910 grants to R.P.

## Author contributions

R.P. and A.H.T. designed the study; Y.S. performed the experiments; A.H.T. and R.P. analyzed the data; R.P. and A.H.T. wrote the manuscript.

## Additional information

**Competing interests:** The authors declare no competing interests.

