## [Peer Review File · Nature Communications]

Reviewers' comments:

Reviewer #1 (Remarks to the Author):

Intertrial intervals (ITIs) carry information that can be used to predict the onset of a following trial, and therefore to form better models of the environment. Typically ITIs are on much longer timescales than inter-stimulus intervals. How the brain represents such long intervals is an outstanding question. In the current manuscript, the authors hypothesized that the ITI is represented in the amygdala.

The authors recorded from dACC and amygdala neurons during a tone-aversive odour conditioning task. Tones and odour presentation were locked to the breathing cycle of the animal. The mean ITI was ~38 seconds, but varied from trial to trial. No behavioural changes during the ITI were observed. However the length of the ITI correlated with both an increased CR and a higher learning rate.

The authors found that 50% of amygdala and dACC neurons had significantly modulated their firing rate during 20% or more of the ITI. Furthermore, the depth of modulation increases with the number of trials, suggesting it is related to learning of the task. The authors hypothesized that the ITI activity in the amygdala and dACC represent an expected ITI length, and therefore hypothesized that variations in ITI length can update the ITI representation from trial to trial. In around 15% of amygdala and dACC neurons, the change in center-of-mass of ITI modulation correlated significantly with the length of the previous ITI.

Despite similar proportions of neurons in the amygdala and dACC with ITI tuning, it appeared that they predominantly are tuned to different periods of the ITI. Amygdala neurons preferentially were tuned to the middle of the ITI, while dACC neurons signaled either early or late ITI. Significant cross-correlations existed between 41% amygdala and dACC neurons pairs, and the significant cross correlations peaked at early and late ITI periods. PCA on the distribution of cross-correlations over the ITI period revealed three clusters. There are three types of dACC-amygdala cell pairs which synchronize during specific segments of the ITI. This suggests that communication between dACC and amygdala contributes to the different peak distributions of amygdala and dACC ITI tunings.

Finally, the authors reasoned that if the dACC and amygdala actually carried reliable representations of the ITI, then they should be able to predict the length of the ITI from the neural data alone. They found that the particular linear decoder they used was able to recreate the ITI from the neural data within a reasonable error, suggesting that these representations can be used by downstream regions to modulate behaviour.

Overall this is a very strong manuscript exploring an interesting topic from one of the leading labs in the field. The question is novel and of broad interest to the scientific community.

I do however, have two small concerns.

1. In Figure 6a, the authors claim that dACC has bimodal distribution of temporal tuning, while amygdala is unimodal. They tested statistically the bimodality of dACC neurons, but not amygdala neurons. It appears by eye that the amygdala distribution might also be bimodal, although less separated than the dACC.

2. The authors claim that the width of these peaks is similar. Again, by eye, it appears as if the amygdala peak width is considerably larger. In the text they claim that they are not significantly different, with a p-value greater than 0.01. Either this is a typo, or they have changed their significance criterion. I do not think that downstream analyses critically depend on this result, but I do think this is something the authors should address.

Reviewer #2 (Remarks to the Author):

Taub AH et al., Long time-scales in primate amygdala neurons support aversive learning

The authors present data from experiments in which they recorded the activity of single neurons in the amygdala and dorsal ACC, while animals underwent a Pavlovian aversive conditioning paradigm. A tone CS was followed by an aversive odor. In some sessions they also examined appetitive conditioning. The inter-trial-interval length was randomized. The authors found that the length of the ITI affected the learning, at least 1 trial into the future. Further, they found that both the amygdala and ACC neurons had time-varying firing rates during the ITI, and this firing rate was related to the learning, and depended on the previous trial's ITI. In a final analysis they found significant cross correlations between ITI activity between the amygdala and ACC.

This paper addressed the question of the relation between the length of the ITI, learning, and firing rate across the amygdala/dACC circuitry. The experiment was well carried out, and the data is well analyzed and clearly presented. I have a few comments on details, and one general comment on what the authors think the neural signal is representing.

Comments

1. My main comment has to do with the interpretation of the physiology result. The authors interpret the result as a correlate of learning. However, the activity may represent anticipation of the aversive event that will be cued by the CS. Anticipation would also depend on the learning, and reflect previous trial outcomes, etc., since the level of anticipation would relate to how well the association has been learned. It may be difficult to disentangle these, but some discussion of this would be useful.
2. Have the authors considered using a hazard rate function, during the ITI to analyze the data? The ITI is of variable length, so this induces a hazard rate on when the cue will appear. Some of the activity may be related to this function.
3. The authors mention that the prediction of elapsed time by the population is, "... an underestimate because we assume independence...". However, the true error is probably larger. Independence usually increases information.
4. Fig. 1C should have a scale bar for time.
5. I could not work out the multiple colors in Fig. 3C. Also, it says 2 cycles are shown, but it is not completely clear how this is shown. Maybe a wide US wedge could be used to show the range of the second US that occurs on the bottom of each polar plot? These plots were a bit confusing.
6. In the results it says that the ITI firing rate increase with acquisition. However, in Fig. 4B it is decreasing across trials of learning. I suppose the authors are referring to the fact that it increases relative to habituation? But across the acquisition trials it seems to be decreasing. This could be explained more clearly in the results.

Reviewer #3 (Remarks to the Author):

The authors used a tone-odor conditioning task with long inter-trial intervals to study the behavioral and neuronal effects of different ITIs. They show that longer ISIs result in strong conditioning, a well known effect in conditioning literature. What is new here is that both amygdala

and ACC cells represent "time elapsed" during the ITI. The authors posit that this representation underlies the role of ISI length in conditioning.

It has long been recognized that ISIs serve an important role in learning, but the underlying representations and neural structures remain unclear. This is thus an important topic to study and the data are interesting and novel.

While this is a very interesting study, it was difficult for me to digest and assess how robust the claims are. In particular, while the early parts clearly show the behavioral effects and basic cellular observation (Figs 1-3, 8), some of the later parts (Fig 4,5,6,7) are less convincing, difficult to understand and seem to be little more than a random set of analysis tangentially relevant to the main claim.

Why not clearly define the preferred time of a neuron (like in Fig 3) and then see whether this preferred time of a such identified neuron changes as a function of learning, expectation, history etc? Instead, a variety of complex correlations are used, leaving it unclear to me how this is related to the types of cells shown in Fig 3. Overall, the paper does not presently well support that this representation of time is related to learning (see my major point #1). But having said that, it does show that a representation of time does exist in amygdala/ACC during ITIs in conditioning, which is novel and very interesting. But the claim in the title is that this representation relates to learning, which I cannot see how that is supported.

Major issues:

1. My most important issue is that it is unclear to me how the authors can claim that this neural representation of time is related to conditioning. The strength of conditioning clearly changes as a function of ITI, as do the amygdala/ACC cells that code for "elapsed time in an ITI" (as the decoding nicely shows). But what shows that these two are related? Some kind of trial-by-trial relationship between firing rates during ITI and strength of conditioning would be required, for example keeping one variable constant (such as the ITI), variability in timing representation predicts extent to which CR strengthens? At present, I cannot see how this principle argument of the paper is supported. The only argument is that it isn't present during extinction, but that does not show that it plays a role in learning.

2. What is missing is rasters of amygdala "time cells" – to see if the response is stereotypical across trials and to see whether this is a robust phenomena.

2. What is the relationship of these findings to the well described "time cells" in the hippocampus? These cells "tile" space in a sequential manner much like the cells described here. While briefly touched upon in discussion, I wonder whether the authors believe that this is the same or different phenomena. If it is the same, there should be some time cells that only fire in the "long trials" (i.e. longer than the mean ITI), when a trial exceeds this. One limitation of the analysis here is that only the data up to the mean ITI is included, leaving it unclear what happens beyond this.

4. Despite my best attempts I'm unable to decipher what argument Fig 5 supports. If I understand correctly, this shows that "time cells" change their preferred time of firing as a function of the length of the previous ITI. But how this supports a model of "expectation" is unclear to me. Presumably, the monkeys expectation is the mean ITI. But whether this expectation is violated or not can only be evaluated once the next trial starts (which provides a prediction error).

3. Fig 6 – while the tiling shown is nice, it is unclear to me to what degree the nice tiling is by selection and whether the differences between the areas are significant.

4. How reproducible are the effects across monkeys?

Reviewer #1

Overall this is a very strong manuscript exploring an interesting topic from one of the leading labs in the field. The question is novel and of broad interest to the scientific community.

I do however, have two small concerns.

We thank the reviewer for this positive evaluation.

1. In Figure 6a, the authors claim that dACC has bimodal distribution of temporal tuning, while amygdala is unimodal. They tested statistically the bimodality of dACC neurons, but not amygdala neurons. It appears by eye that the amygdala distribution might also be bimodal, although less separated than the dACC.

We agree. Our statement was more qualitative than quantitative to emphasize the fact that amygdala neurons have peaks more in the middle of the ITI. The statistical test the reviewer refers to compares the two distributions (amygdala vs. ACC) and shows they are significantly different– it is not a test for bimodality for ACC (or Amygdala) – we did not perform such direct test. We apologize for the misunderstanding. We revised the text to make it clear.

Due to requests from the other reviewers, we re-calculated the peaks and their location for an extended period during the ITI (including early and late responses as well).

We therefore revised Fig.6 completely, and the results show that our original finding is robust, but also extends it. To address the issue more directly, we fitted gaussians to the peak distributions, and indeed find that both regions have two main peaks: one early and one later. Yet the main density of amygdala late peaks is in the middle of the ITI (as described in the original version), and the main density of ACC late peaks is later towards its end (as described in the original version). Please see the revised figure 6, the additional analyses, and the revised text.

2. The authors claim that the width of these peaks is similar. Again, by eye, it appears as if the amygdala peak width is considerably larger. In the text they claim that they are not significantly different, with a pvalue greater than 0.01. Either this is a typo, or they have changed their significance criterion. I do not think that downstream analyses critically depend on this result, but I do think this is something the authors should address.

We believe there is misunderstanding and we did not explain properly. There was indeed no difference in the mean tuning-width. Perhaps the reviewer refers to the width of the distribution of peaks? This is indeed different, as shown by the new fits to gaussians (revised fig.6), yet the average width of single-units tuning is not different between amygdala and ACC (as shown in more detail in the revised fig.6, additional analyses, and related text). We now show this in addition as a function of trials in learning (Fig.6) and separately for the two animals (supp.fig.5), strengthening the finding

that the tuning width of individual cells is similar (on average). We revised to clarify and added the new analyses and aforementioned figures.

Reviewer #2

...The experiment was well carried out, and the data is well analyzed and clearly presented. I have a few comments on details, and one general comment on what the authors think the neural signal is representing.

We thank the reviewer for the positive evaluation.

1. My main comment has to do with the interpretation of the physiology result. The authors interpret the result as a correlate of learning. However, the activity may represent anticipation of the aversive event that will be cued by the CS. Anticipation would also depend on the learning, and reflect previous trial outcomes, etc., since the level of anticipation would relate to how well the association has been learned. It may be difficult to disentangle these, but some discussion of this would be useful.

In short, we agree and added a sentence in the discussion to mention this.

Yet I want to clarify that we do not see this as ‘a correlate of learning’, but as information that exists during learning and can be used to aid proper learning. In addition, I want differentiate two things: what we show is that learning induced a neural representation during the ITI. This representation can be used to: 1. Correct anticipation of the next CS/US; 2. Contribute to the rate and strength of learning. In behavior, we find evidence for #2 (see Fig.2), yet no concrete evidence for #1 (see Fig.1). We also find a direct link between neural representation during the ITI and #2 (new analyses shown in fig.5 and text). This is why we think the representation of the ITI duration serves here more as a mediator aiding learning, rather than pure anticipation of the CS-US to come.

However, if the reviewer refers to US anticipation alone (i.e. excluding the CS), then indeed these two interpretations cannot be disentangled here completely (they are not mutually exclusive); because if US anticipation is triggered by the CS only, then the CR is the only measure/correlate of learning. In other words, if CS anticipation is in neural response only and behavior waits until the CS to show US anticipation, then it is indeed inline with our data and we cannot differentiate the two interpretations (frankly, I think they are similar in nature in the sense that it is the same information that can be used for both).

We do find anticipation during the ITI in neural response though, after implementing the novel analyses in response to point #2 by this reviewer– the hazard rate (see next answer). The fact that there is no behavioral correlate during the ITI (atleast in the parameters we quantified) could simply mean that since the CS is a complete predictor of the US, a behavioral output is not required (inline with classical RL). The anticipation and its violence in neural activity can be used to alter the change in CR, and hence modulate directly the learning rate.

In addition, In response to the first concern of reviewer 3, we added more analyses that show a link between the neural activity representing ITI-duration on a trial-by-trial basis and the rate of learning. Therefore, our main interpretation is that representation of ITI duration contributes to learning-rate/strength.

Notice also that even in our original introduction and discussion we mentioned that “ In associative learning, the passage of time between trials – the inter-trial-interval (ITI), can potentially serve as a cue of trial expectation. The time that passes from the offset of one trial carries information about the onset of the next trial, and If this time can be internally represented and kept, it can aid to form higher-order representations of the environment. “ and “can support formation and maintenance of temporal contingencies not only when they are cued

explicitly by external stimuli, but also when embedded in the temporal structure of the task”, and we also discuss models of ‘bridging’ and formation of association between the ITI and the CS and even the US (see a full paragraph in the discussion).

Yet in light of how we understand the reviewer comment (direct expectation of the US, independent of the CS), we also added this more specific interpretation briefly in the discussion.

2. Have the authors considered using a hazard rate function, during the ITI to analyze the data? The ITI is of variable length, so this induces a hazard rate on when the cue will appear. Some of the activity may be related to this function.

This is an excellent suggestion and we therefore performed these analyses which resulted in new figure 9 (most relevant here is simply the cumulative $p(\text{CS})$ as a function of time in the ITI, computed empirically from the actual distribution of the ITI lengths). Please see the figure and the added results for details. In short, we indeed find that a proportion of the cells seem to have activity that correlates with the hazard rate. It therefore strengthens the results because it suggests that a significant proportion of cells provide expectation of the coming CS (or the US, see previous answer per the reviewer suggestion), even when faced with such long and variable timescales during the ITI. To be able to have such an anticipatory activity, the network must maintain long time scales at the order of dozens of seconds, as we show here.

3. The authors mention that the prediction of elapsed time by the population is, “... an underestimate because we assume independence...”. However, the true error is probably larger. Independence usually increases information.

Yes, ofcourse, we mean the same obviously. It is a language mistake on our side. We say “...few (<4) seconds accuracy... is an under-estimate...”, and we meant it is probably more than 4 seconds (hence the ‘underestimate’...). We revised to proper English. Thanks.

4. Fig. 1C should have a scale bar for time.

Added. Notice this is a 1-day example for illustration and the exact mean (red line) is mentioned in the legend (32 sec that day).

5. I could not work out the multiple colors in Fig. 3C. Also, it says 2 cycles are shown, but it is not completely clear how this is shown. Maybe a wide US wedge could be used to show the range of the second US that occurs on the bottom of each polar plot? These plots were a bit confusing.

We know it is a non-traditional plot yet presenting two consecutive ITI of such long and variable periods (dozens of seconds) looks extremely weird and hard to read in standard PSTH presentation (see the new added example raster/PSTH in Supp.Figs. of only one ITI - even one ITI is hard to interpret for such long period in horizontal presentation). It also allows to see phase of modulation with such alignment and presentation.

We revised the text to provide clearer description and I hope it is better now.

To clarify: each CS-US is on top (at 90 degrees), with the ITI that comes before it on the left half-circle (i.e. ends at the CS), and the ITI that comes after it on the right half-circle (i.e. starts after the US). The green-red colormap matches the vector size (the instantaneous firing-rate), for presentation purposes only.

The US is marked in a wide purple wedge. Adding a second US at the bottom will require a US on the bottom left side and a CS on the right bottom side (and a very wide wedge b/c of wide ITI distribution, and deciding if to add the left side CS also with a wedge and the right side US also with a wedge), so I fear it will be even more confusing... Notice we do say ‘mean ITI’ in each bottom side.

6. In the results it says that the ITI firing rate increase with acquisition. However, in Fig. 4B it is decreasing across trials of learning. I suppose the authors are referring to the fact that it increases relative to habituation? But across the acquisition trials it seems to be decreasing. This could be explained more clearly in the results.

Thanks. This is indeed what we meant. We revised in the results and the figure legend to make it clearer. We initially thought this decrease can be a correlate of learning (less representation is required once learning reaches a plateau, and/or with reduction in the prediction-error), but the effect was too weak to draw strong conclusions.

Reviewer #3

....But having said that, it does show that a representation of time does exist in amygdala/ACC during ITIs in conditioning, which is novel and very interesting. But the claim in the title is that this representation relates to learning, which I cannot see how that is supported.

We thank the reviewer for the constructive criticism and suggestions of how to address it. We hope our new analyses address the main concerns and provide stronger support for our findings, as well as clarify the interpretation.

1. My most important issue is that it is unclear to me how the authors can claim that this neural representation of time is related to conditioning. The strength of conditioning clearly changes as a function of ITI, as do the amygdala/ACC cells that code for "elapsed time in an ITI" (as the decoding nicely shows). But what shows that these two are related? Some kind of trial-by-trial relationship between firing rates during ITI and strength of conditioning would be required, for example keeping one variable constant (such as the ITI), variability in timing representation predicts extent to which CR strengthens? At present, I cannot see how this principle argument of the paper is supported. The only argument is that it isn't present during extinction, but that does not show that it plays a role in learning.

To address this, we performed a novel analysis, correlating between the change in CR (behavioral learning rate) and the center-of-mass of the neural activity during the ITI.

The idea is simple: as the reviewer suggests, this offers a trial-by-trial relationship between the relevant neural factor during the ITI – “variability in timing representation”, and “extent to which CR strengthens”. Although we do not hold the ITI constant per-se (because we do not have enough statistical power), it neglects the ITI length itself, essentially averaging over it.

The new results, added to Fig.5 and Supp.Fig.2 and to the revised text, show a significant trial-by-trial correlation between the two measures, and provide a direct relationship between the neural temporal representation of the ITI and the rate of learning. The relationship was significant in both regions, but with stronger correlation in the amygdala (see Supp.Fig.2). In addition, firing rate from the whole ITI did not show such relationship (data not shown unless the reviewer would ask for it to be included), strengthening the notion that it is the temporal representation (represented by center-of-mass) rather than overall activity.

We hope this is satisfactory. It adds to the already existing indirect evidence supporting this, i.e. the correlations presented between the ITI length and the rate of learning (Fig.2), and between the ITI length and the center-of-mass of single-cell neural activity (Fig.5). Together, the three correlations (not independent, but also not redundant) provide strong support that there is indeed a relationship between representation of ITI duration in single-cells and learning rate.

2. What is missing is rasters of amygdala "time cells" – to see if the response is stereotypical across trials and to see whether this is a robust phenomenon.

We added few rasters as example to the new Supp.Fig.4 and to the new Fig.9

However, rasters over dozens of seconds w/o any external stimuli are very hard to show reliability in timing, and very few studies if at all show such rasters (place-cells, for comparison, do have external trigger – the place). We are therefore very careful of calling them ‘Time-cells’, as they are indeed less reliable and rarely show locked-in-time activity over dozens of seconds in raster presentation. This is also reflected in the width of tuning and in the error when predicting time from population responses. The strength is in the population distributed over 2 regions and the sparse representation within the ITI, not in single-cell tuning reliability. This is an important aspect of our manuscript and its interpretation.

3. What is the relationship of these findings to the well described "time cells" in the hippocampus? These cells "tile" space in a sequential manner much like the cells described here. While briefly touched upon in discussion, I wonder whether the authors believe that this is the same or different phenomena. If it is the same, there should be some time cells that only fire in the "long trials" (i.e. longer than the mean ITI), when a trial exceeds this. One limitation of the analysis here is that only the data up to the mean ITI is included, leaving it unclear what happens beyond this.

As the reviewer notes, we are careful not to claim these are similar to ‘time-cells’, and we do not know if it is the same mechanism. We do suggest in the discussion the information can come from hippocampal time-cells (and slightly revised the references), but we obviously cannot know this here. It certainly looks less reliable than classical ‘time-cells’ (see previous response).

Due to statistics, we cannot reliably identify cells that have a peak-modulation after the mean-ITI, mainly because there only few trials to identify a real peak there. We did however perform an additional analysis of hazard-rate function, which allowed us to identify cells that have gradual increase in activity after the mean ITI (see response to point#2 reviewer#2 and the new Fig.9). At this point we cannot determine if a peak that appears after the mean-ITI is a real peak or an anticipation towards the CS, mainly due to statistical limitations. We supply the proportion of cells that were classified both as having a peak and as anticipatory activity in the new Fig.9, and so allow both interpretations for this small population of cells.

4. Despite my best attempts I'm unable to decipher what argument Fig 5 supports. If I understand correctly, this shows that "time cells" change their preferred time of firing as a function of the length of the previous ITI. But how this supports a model of "expectation" is unclear to me. Presumably, the monkeys expectation is the mean ITI. But whether this expectation is violated or not can only be evaluated once the next trial starts (which provides a prediction error).

This is indeed what it shows (“change their preferred time of firing as a function of the length of the previous ITI”). The idea is that if cells indeed hold a representation of the time between events and it affects learning, then this representation should be updated if the previous trial was longer or shorter. It provides demonstration that cells care about the ITI length dynamically during learning, and not only signal a static parameter (even if learned), or simply time from the US. As such, it also provides an additional control for US-related activity or any other potential external stimuli. This argument is now much stronger following our new analyses in response to concern #1 by this reviewer.

Most cells do not show activity related to anticipation per-se (but some do, see the new Fig.9), and behavior does not show it as well (Fig.1), and unfortunately, we cannot observe violations (prediction-error) due to the CS response (this is a limitation of the study). Please also see our response to concern #1 for reviewer #2 about expectation.

If I understand correctly, the reviewer is right that violation of ITI duration can be observed only when the CS appears, but our interpretation and what we show is that the previous ITI duration affects the next ITI neural representation, and that ITI duration affects the following CR (new analyses) – this indeed can be driven by a violation signal ‘hidden/added’ in the response to the CS.

Unfortunately, we cannot in this study identify such signal that occurs at a specific point in time (i.e. when the ITI ends). This is because of 2 confounds: for shorter than average ITIs, the CS-evoked activity masks any temporal-based prediction-error; and for longer than average ITIs, the lower number of trials (see hazard analysis in the new Fig.9) and the very ‘loose’ representation of time in such long time-scale (dozens of seconds), probably prevent us from observing a reliable ‘disappointment’ signal (as in classical RL / TD-error studies that observe such signals when expecting rewards at 0-3 seconds delay). We looked carefully again but identifying reliably a prediction-error signal at >30 seconds provided very weak statistical results. To conclude, although we do believe violation of expected long ITI probably occurs to drive the correlations we present, we could not identify such signal, atleast when recording only few cells simultaneously.

3. Fig 6 – while the tiling shown is nice, it is unclear to me to what degree the nice tiling is by selection and whether the differences between the areas are significant.

I am not sure what is meant by ‘by selection’. The result is robust when taking peaks only from cells that significantly modulated their activity during the ITI, and when taking peaks from all cells. Nevertheless, we re-calculated the distributions in a slightly different way, allowing all peaks during the ITI to enter (one peak per cell ofcourse). The revised Fig.6 shows that the original finding is robust and distributions are different between regions. It now shows that although both regions had a large number of peaks early-on after the US, the second peak was differential across regions - the amygdala is more prone to have peaks in the middle of the ITI, and the ACC is more prone to have peaks late in the ITI.

If the reviewer means that it might occur by sampling (i.e. that if we sampled enough neurons we would observe homogenous distribution in both regions), this is ofcourse always a possibility. I believe we have enough cells in the study to suggest the difference is real (it is significant), and moreover, the independent cross-correlation analyses (Fig.7), again shows non-homogenous cross-regional synchrony along the ITI.

4. How reproducible are the effects across monkeys?

We repeated several main analyses separately for the two animals and report them in the newly added supp.fig.5.

REVIEWERS' COMMENTS:

Reviewer #1 (Remarks to the Author):

The authors have addressed my comments. Recommend accept.

Reviewer #2 (Remarks to the Author):

The authors have addressed all of my concerns. I have no further comments.

Reviewer #3 (Remarks to the Author):

The authors were very responsive to the issues I raised. The new analysis and replies in the rebuttal provided address my issues fully. Despite the authors reservations, I believe the rasters shown in Fig S4 are useful to see (and show the effect). I am thus supportive of publication of this manuscript in this form after addressing the following minor concerns (which do not require re-review by me).

Minor issues:

- Please clarify in methods how exactly center of mass was determined. Also, the units of the x axis in the center of mass plots is unclear (Fig 5C, S2). Is it seconds?
- The new trial-by-trial analysis shown in Fig 5 is very interesting, but what I'm missing is an interpretation of what this means (in discussion). i.e. why do the authors think this change in where the neuron fires relates to learning? Similarly, the correlation of center-of-mass with previous trial ITI length presumably means that neurons update their expectation of how long the ITI will be, and this is reflected in shifts of where the center-of-mass is located? A few sentences that place this interesting result into context of the broader literature would be great.

Reviewer #3 (Remarks to the Author):

- Please clarify in methods how exactly center of mass was determined. Also, the units of the x axis in the center of mass plots is unclear (Fig 5C, S2). Is it seconds?

Indeed, it is seconds. We added the explicit formula to the Methods section.

- The new trial-by-trial analysis shown in Fig 5 is very interesting, but what i'm missing is an interpretation of what this means (in discussion). i.e. why do the authors think this change in where the neuron fires relates to learning? Similarly, the correlation of center-of-mass with previous trial ITI length presumably means that neurons update their expectation of how long the ITI will be, and this is reflected in shifts of where the center-of-mass is located? A few sentences that place this interesting result into context of the broader literature would be great.

We revised the discussion to refer explicitly to this interesting result.